# *Bacillus amyloliquefaciens* Lysate Ameliorates Photoaging of Human Skin Fibroblasts through NRF2/KEAP1 and TGF-β/SMAD Signaling Pathways

**Yongtao Zhang** [1,2,†], **Jingsha Zhao** [1,2,†], **Yanbing Jiang** [1,2], **Dongdong Wang** [1,2], **Dan Zhao** [1,2], **Changtao Wang** [1,2] **and Meng Li** [1,2,*]

1   Beijing Key Laboratory of Plant Resource Research and Development, College of Chemistry and Materials Engineering, Beijing Technology and Business University, Beijing 100048, China
2   Institute of Cosmetic Regulatory Science, Beijing Technology and Business University, Beijing 100048, China
*   Correspondence: limeng@btbu.edu.cn; Tel.: +86-13426015179
†   These authors contributed equally to this work.

**Abstract:** More and more research in dermatology and cosmetic science is devoted to the development and application of postbiotic raw materials. In order to explore the anti-photoaging efficacy and application prospect of *Bacillus amyloliquefaciens* lysate (BAL1) on the skin, we used 16 J/cm$^2$ UVA stimulation of human embryonic fibroblasts (CCC-ESF-1) to establish a UVA photodamage model to investigate the anti-photoaging efficacy of BAL1 and its mechanism of action. In this study, we found that BAL1 activated the transcription of downstream antioxidant enzyme genes mainly by promoting the nuclear displacement of NF-E2-related factor 2 (Nrf2) within CCC-ESF-1, thus increasing the antioxidant capacity of antioxidant enzymes to scavenge excessive reactive oxygen species in cells. Meanwhile, BAL1 promoted intracellular TGF-β/Smad signaling pathway and reduced matrix metalloproteinase expression to alleviate the degradation of extracellular matrix. In conclusion, the results of this study demonstrate the potential benefit of BAL1 in protecting the skin against UVA-induced photoaging and highlight the potential of BAL1 in skin photoprotection.

**Keywords:** *Bacillus amyloliquefaciens*; postbiotics; lysate; Nrf2; antioxidant; anti-photoaging

## 1. Introduction

Postbiotics are another new concept after prebiotics and probiotics, mainly refers to microbial metabolites that are beneficial to the host, the composition of the bacterium and its inactivated bacterium itself [1]. Postbiotics have the advantage of greater stability, absorption, and metabolism than live bacteria, which is very helpful in ensuring effectiveness throughout storage and use [2,3]. The most widely used categories of postbiotics mainly include cell-free fermentation filtrate, bacteriostats, polysaccharides, and antioxidant peptides [3,4], antibacterial peptides, antioxidant enzymes, and other active ingredients. With broad-spectrum antimicrobial, antioxidant, and immunomodulatory activities, postbiotics also affect specific physiological responses through interactions with different molecules or receptors or indirect mechanisms involving microbiota homeostasis or host metabolic and signaling pathways [5,6]. For skin problems, the research and application of postbiotics have an excellent development trend and is a current research hotspot. For example, previous studies in our laboratory explored that SF-1-ECPs (derived from *Saccharomycopsis fibuligera*) have anti-photoaging and anti-oxidative stress effects [7]. Muhammed Majeed et al. [8,9] found that LactoSpore® (derived from *Bacillus coagulans*) has strong antioxidant, anti-photoaging, and anti-inflammatory effects.

The research on probiotics and postbiotics in skincare has become popular in recent years and they are highly effective natural products. However, related studies are currently scarce despite good research and application prospects. *Bacillus amyloliquefaciens* is an

FDA-recognized probiotic with a broad spectrum of antibacterial, immunomodulatory, anti-inflammatory, and antioxidant activity [10]. However, the application of *B. amylolyticus* is mainly concentrated in the fields of agriculture and animal husbandry, and there are few cases of application in food, medicine, and cosmetics. Recently, L S Safronova et al. [11] explored the cell-free extracts of *B. amyloliquefaciens* ssp. *plantarum* containing gallic, 4-hydroxyphenylacetic, caffeic, syringic, p- coumaric, trans-ferulic, and trans-cinnamic acids, which effectively protect cells from oxidative damage. Minkyoung Kang et al. [12] found that *B. amyloliquefaciens* can significantly prolong the lifespan of *C. elegans*, and verified by RAW264.7 that it can effectively reduce the inflammatory response. Yeqi Zhao et al. [13] found that the Nrf2/OH-1 pathway could be effectively activated to alleviate gastrointestinal oxidative damage by feeding *B. amyloliquefaciens* suspension to aflatoxin B1-induced mice. In addition, Md. Saifur Rahman et al. [14] isolated and purified an antioxidant peptide YD1 from *B. amyloliquefaciens* CBSYD1 strain and showed that YD1 could enhance the transcription and translation of Nrf2 in cell RAW 264.7 and improve the level of antioxidant enzyme activity to reduce the level of reactive oxygen species. This fully demonstrates the value and significance of understanding *B. amyloliquefaciens* being further developed and applied, but no studies have been conducted to investigate the efficacy of *B. amyloliquefacien* lysate in skin care or other areas.

In this study, we isolated a strain of *B. amyloliquefaciens* BA-1 strain from rice wine koji, and preliminary investigations revealed that *B. amyloliquefaciens* BA-1 strain lysate (BAL1) has a strong ability to scavenge free radicals. In this study, to further investigate the antioxidant activity and anti-aging activity of BAL1, in vitro antioxidant, cellular experiments and cellular, molecular experiments were conducted to explore the application value of BAL1 as a functional ingredient in cosmetics.

## 2. Materials and Methods

### 2.1. Materials

*B. amyloliquefaciens* BA-1 strain was isolated from rice wine koji. Human embryonic skin fibroblast CCC-ESF-1 was purchased from Peking Union Medical College Hospital, 1,1-diphenyl-2-trinitrophenylhydrazine, ascorbic acid was purchased from Beijing Biodee Biotechnology (Beijing, China) Co., Ltd.; DMEM medium, Fetal Bovine Serum, penicillin-streptomycin, and 0.25% trypsin (including EDTA) were purchased from Thermo Fisher Scientific (Shanghai, China) Co., Ltd. Phosphate Buffered Saline (1 × PBS), Cell Counting Kit-8, UEIris II RT-PCR System for First-Strand cDNA Synthesis and Fast Super EvaGreen® qPCR Master Mix were purchased from Biorigin (Beijing, China) inc. Western and IP Cell lysis Buffer, Phenylmethanesulfonyl fluoride (PMSF), Trizol, and Reactive Oxygen Species Assay Kit were purchased from Biyuntian Biotechnology (Shanghai, China) Co., Ltd.

### 2.2. Culture and Extract

*B. amyloliquefaciens* BA-1 strain was isolated from rice wine koji. Strain identification was performed and analyzed by Shanghai Majorbio Bio-pharm Technology Co., Ltd. The spliced sequences after sequencing were aligned with NCBI. The nucleotide sequence of this strain has been deposited in GenBank ID OK217228.

*B. amyloliquefaciens* BA-1 strain was cultured with LB medium in a constant temperature shaker at 37 °C and 180 rpm for 1d. The BA-1 suspension was centrifuged at 10,000 rpm to obtain a bacterial pellet. The obtained bacterial precipitates were mixed well with Tris-HCl buffer (pH = 7.4), supplemented with 1:1000 protease inhibitor (PMSF), and broken by cell ultrasonic crusher (JY92-IIIN, Ningbo Xinzhi Biotechnology (Ningbo, China) Co., Ltd.) under ice bath with machine settings: working frequency 200 W, working time 20 min (ultrasonic 10 s, intermittent 10 s). Hence, 10,000 rpm centrifugation was performed to extract the supernatant, and the supernatant obtained was lyophilized to obtain BAL1.

### 2.3. LC-MS/MS

LC-MS/MS analysis was performed on a TripleTOF 5600 System (SCIEX, Foster City, CA, USA) coupled on-line to an Ultra Plus NanoLC 2D HPLC system (Eksigent Technologies, Dublin, CA, USA). Lyophilized peptides were dissolved in solvent A (2% CAN and 0.1% FA in HPLC grade water) and centrifugated for 10 min at $17,000 \times g$. The peptide mixture was transferred to a 2 cm self-packed trap column (100-μm inner diameter, 3 μm resin, ReproSil-Pur C18-AQ, Dr Maisch GmbH) at 3 μL/min in 10 min. After loading and washing, peptides were separated onto a 12 cm analytical C18 column (150-μm-inner-diameter, 1.9 μm resin, ReproSil-Pur C18-AQ, Dr Maisch GmbH) via a 55 min non-linear gradient from 8% to 90% solvent B (0.1% formic acid in acetonitrile) at 600 nL/min. The gradients were as follows: 0–8 min, 8 to 25% B (acetonitrile, 0.1% formic acid), followed by 8 min linear gradient to 35% and 9 min linear gradient to 90%. The eluted peptides were ionized by nanoliter electrospray into a tandem mass spectrometer TripleTOF 5600 (SCIEX, Foster City, CA, USA). The ion spray voltage was set at 2300, ion source temperature at 150 °C, gas curtain gas at 35 psi, nebulizer gas at 20 psi, primary mass spectrometry (MS) scan range 350–1500 (m/z), accumulation time 0.25 s, secondary (MS/MS) in high sensitivity mode, scan range 100–1500 (m/z), accumulation time 0.05 s. One MS map was used for up to. The top 40 parent ions of signal intensity were selected for cascade scanning.

The files generated by mass spectrometry were identified by importing into Protein-Pilot 5.0 software (AB SCIEX). The database search parameters are as follows: Sample Type: Sample Type = Identification; Digestion = Trypsin; Instrument = TripleTOF 5600; Database = *Bacillus amyloliquefaciens* protein database downloaded from the uniprot website (https://www.uniprot.org/ (accessed on 15 April 2022)). Protein level false positive rate (FDR) < 1%.

### 2.4. Free Radical Scavenging Ability

The 40 mg/mL BAL1 was used as the initial concentration to dilute 6 concentration gradients by a two-fold dilution, and the same concentration of ASCORBIC ACID was used as a positive control. Hence, 100 μL of DPPH and 100 μL of sample were added to a 96-well microtiter plate and reacted for 30 min, and $OD_{517}$ was measured by microplate reader (MULTISKAN FC, Thermo Fisher Scientific (Shanghai, China) Co., Ltd.). Set the sample group (T), blank group (C) in the experiment.

Mix 300 μL of 6 mM ferrous sulfate and 300 μL of 6 mM $H_2O_2$. Then, mix with 300 μL of 6 mM ethanol-salicylic acid solution and incubate at 37 °C for 15 min. Add 100 μL of distilled water as blank group (C), and add 100 μL of sample as sample group (T). Continue to incubate at 37 °C for 15 min. Add the reaction solution to each well of a 96-well plate, and measure $OD_{510}$.

Mix 7.4 mM ABTS and 2.6 mM $K_2S_2O_8$ evenly, incubate in the dark at room temperature for 12 h, and dilute with distilled water to $OD_{734}$ at 0.7 to obtain ABTS working solution. Add 200 μL of ABTS working solution to each well of the 96-well plate, add 10 μL of water as blank group (C), add 10 μL of sample as sample group (T), react for 4 min, and measure $OD_{734}$.

The formula for the free radical scavenging rate is as follows:

$$\text{Free radical Scavenging rate }(\%) = (1 - \text{T/C}) \times 100\% \tag{1}$$

### 2.5. Cell Viability Assay

CCC-ESF-1 was cultured in cultured in HeraCell $CO_2$ incubator (BPN-50CH(UV), Shanghai Yiheng Scientific Instrument (Shanghai, China) Co., Ltd.) from complete medium DMEM.

Briefly, 100 μL of CCC-ESF-1 suspension ($1 \times 10^5$ cells/mL) was added to 96 microtiter plates and cultured for 12 h. The culture medium was replaced with 100 μL of BAL1 and 100 μL of basal medium for 24 h. Finally, the culture medium was incubated with 10 μL of CCK8 and 100 μL of DMEM for 2 h, and $OD_{450}$ was measured.

In experiments exploring the photoprotective effect of BAL1 on cells, CCC-ESF-1 was cultured in a complete medium for 12 h to adhere. The basal medium containing BAL1 was used to continue to culture for 12 h, and then cells were irradiated with UV cross-linker (UV07-II, Ningbo Xinzhi Biotechnology (Ningbo, China) Co., Ltd.). The total energy was set to 16 J/cm$^2$ at 365 nm, and cell viability was determined after culturing for 6 h.

The formula for cell viability is as follows:

$$Cell\ viability(\%) = 100\% \times (A_s - A_0)/(A_c - A_0) \tag{2}$$

* $A_s$: Sample group, $A_c$: Blank group, $A_0$: Solvent base value

### 2.6. Reactive Oxygen Species Detection

Briefly, 2 mL CCC-ESF-1 suspension ($1 \times 10^5$ cells) was added to the 6-well plate for 12 h culture, and the medium was changed to serum-free medium containing samples for further incubation for 12 h. After cells were irradiated with 365 nm UVA irradiation with a total intensity of 16 J/cm$^2$, incubation was continued for 6 h, before discarding 1 mL of DCFH-DA on each well of the medium and incubating in the dark for 25 min. The 6-well plate was washed with PBS for excess DCFH-DA, and 1 mL of PBS was added to each well to observe the fluorescence state of the cells under a fluorescence microscope. The cells were collected, and a microplate reader was used to detect the fluorescence intensity at the excitation wavelength of 488 nm and the emission wavelength of 525 nm.

### 2.7. Detection of Total Antioxidant Capacity and Antioxidant Enzyme Activity

UVA-induced damage of CCC-ESF-1 was lysed by 200 μL cell lysate (Western IP cell lysate: PMSF = 100:1) for 2 min and collected. To obtain the supernatant, the collected lysed cells were centrifuged at 10,000 rpm and 4 °C for 10 min. Whether BAL1 enhances the total antioxidant capacity and antioxidant enzyme activity of CCC-ESF-1 was explored using the total antioxidant capacity test kit (ABTS method), total SOD activity detection kit (WST-8 method), catalase detection kit, and glutathione peroxidase detection kit (Biyuntian Biotechnology (Shanghai, China) Co., Ltd.).

### 2.8. Enzyme Linked Immunosorbent Assay

The treated cells were extracted with nuclear protein and cytoplasmic protein extraction kit (P0028, Shanghai Biyuntian Biotechnology Co., Ltd.) to extract total cytoplasmic protein and total nuclear protein respectively. The protein content can be quickly and accurately detected by the ELISA method. The protein content was detected according to the operating procedures of HO-1, NOQ1, Nrf2, Keap1, MMP-1 COL-I ELISA kit (CUSABIO, https://www.cusabio.com/ (accessed on 23 October 2021)) and GCLC ELISA kit (Abcam (Shanghai, China) Trading Co., Ltd.).

Add 100 μL of the extracted protein solution to each well and incubate at 37 °C for 2 h. After discarding the protein solution, add 100 μL of Biotin-antibody to each well and continue to incubate for 1 h. After discarding the Biotin-antibody, add 100 μL of HRP-avidin to each well and continue to incubate for 1 h. Finally, after discarding HRP-avidin, add 90 μL of TMB to each well to develop color for 25 min and stop the color development. Measure the absorbance at 450 nm and calculate the corresponding concentration. Each step needs to be washed twice with washing solution.

### 2.9. qRT-PCR

The total RNA of CCC-ESF-1 was extracted with Trizol. The regulation of BAL1 on the Nrf2/Keap1, TGF-β/Smad signaling pathways and other related genes were investigated by qRT-PCR. UEIris II RT-PCR System for First-Strand cDNA Synthesis was used for reverse transcription, Fast Super EvaGreen® qPCR Master Mix was used in qRT-PCR (Thermo Fisher Scientific (Shanghai, China) Co., Ltd.). Search for target genes and design primer sequences on the NCBI (https://www.ncbi.nlm.nih.gov (accessed on 14 February 2022)), as shown in Table S1.

### 2.10. Statistical Analysis

Three or more independent experiments were performed for all experiments and data in our experiments are presented as mean ± standard deviation. Data processing and analysis were performed using GraphPad Prism9 (GraphPad Software, Inc., La Jolla, CA, USA).

One-way ANOVA was used for variance analysis, compare each group with UVA model group, the significant differences were indicated by "*: $p < 0.05$, **: $p < 0.01$, ***: $p < 0.001$".

## 3. Results

### 3.1. LC-MS/MS Analysis

The antioxidant activity of proteins and peptides with antioxidant properties is closely related to their molecular mass size, hydrophobicity, and amino acid composition and sequence, which vary in power/hydrogen supply, hydrophobicity, and metal chelating ability. As measured by LC-MS/MS, BAL1 contains many small molecule proteins and peptides that exhibit strong antioxidant properties in their amino acid sequences and compositions. The protein information aligned from the B. amyloliquefaciens protein database from the uniprot website (https://www.uniprot.org/ (accessed on 15 April 2022)) is shown in Table S2, such as REVERSED Phage tail protein OS (Sequence: VEAVWRSR) and Uncharacterized protein OS (Sequence: LQLVTPLK). The N-terminal Val and Leu have strong hydrophobicity, in addition to the hydrophobicity of Ala, Val, Leu, and Pro contained in amino acid residues, which can promote the solubilization of peptides at the lipid–water interface and thus play a better role in scavenging free radicals [15,16]. Moreover, the power/hydrogen and metal chelating ability of Trp and Glu incubated the ability to scavenge free radicals [17].

### 3.2. Antioxidant Capacity of BAL1

There are many kinds of free radicals (hydroxyl radicals, superoxide anions, nitrogen radicals, etc.) in the cellular system. Each has different chemical and physical characteristics, and a certain antioxidant may have multiple mechanisms in a single system simultaneously. In order to test whether BAL1 has which free radical scavenging ability, we examined the scavenging levels of hydroxyl radicals and nitrogen radicals by BAL1 using the Fenton method, DPPH method, and ABTS method, respectively. The experimental results revealed that BAL1 has a powerful ability to scavenge nitrogen radicals. As shown in Figure 1, the antioxidant capacity of BAL1 showed a dose dependence, with the scavenging efficiency of both DPPH- and ABTS-+ reaching about 90% at 40 mg/mL. In contrast, the scavenging of hydroxyl radicals by BAL1 was weaker than that of nitrogen radicals, with a scavenging rate of 53.6% at 40 mg/mL.

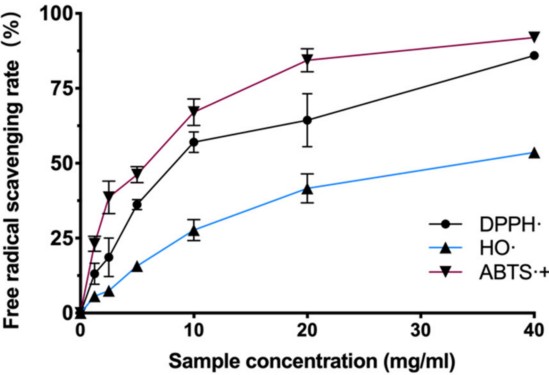

**Figure 1.** Antioxidant ability of BAL1.

### 3.3. Cytotoxicity and Cell Viability

The cytotoxicity of BAL1 against CCC-ESF-1 and the protection of CCC-ESF-1 from UVA damage are shown in Figure 2. BAL1 had low toxicity and was non-lethal to cells in the concentration range of 37.5–300 µg/mL, with survival rates at and above 100%. After incubation of BAL1 at a UVA dose of 16 J/cm$^2$, a significant protective effect of BAL1 at concentrations greater than 150 µg/mL was seen in terms of cell viability ($p < 0.001$), and its survival rate increased from 45.58% in the model group to more than 70%, significantly reducing the mortality of CCC-ESF-1 due to UVA damage. For subsequent experiments, 75.5, 150 and 300 µg/mL were used as concentrations to explore the functionality of BAL1.

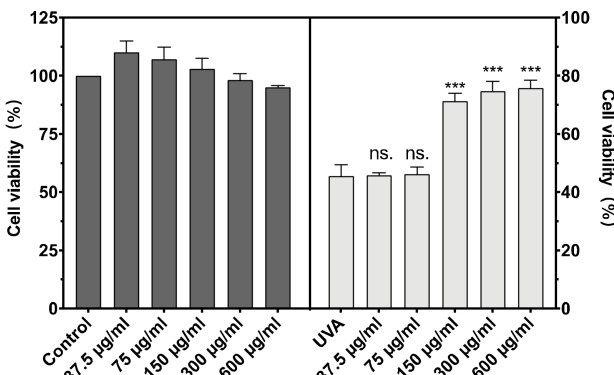

**Figure 2.** Cytotoxicity of BAL1 (Left) and cell survival rate after UVA damage (Right). Difference analysis between sample group and UVA model group: ns.: $p > 0.05$, ***: $p < 0.001$.

### 3.4. The Effect of BAL1 on Cellular Antioxidant Capacity

The reactive oxygen species (ROS) levels and total antioxidant capacity of CCC-ESF-1 are shown in Figure 3. CCC-ESF-1 showed a dramatic increase in intracellular ROS levels under UVA irradiation at 16 J/cm$^2$. Its fluorescence intensity increased from 1203 to 2431 in blank group cells. The corresponding total cellular antioxidant capacity decreased from 1.58 mM Trolox to 0.9382. However, after the cells incubated with BAL1 were stimulated with UVA at 16 J/cm$^2$, the intracellular ROS levels decreased significantly in 150 and 300 µg/mL BAL1-treated cells, and the fluorescence intensity decreased to 1869 and 1784, respectively. The total antioxidant capacity increased to 1.25 and 1.22 mM Trolox, respectively. The ROS levels and the total antioxidant capacity exhibited by BAL1 and ascorbic acid groups were similar, indicating that BAL1 can significantly enhance the antioxidant capacity of cells to slow down UVA damage.

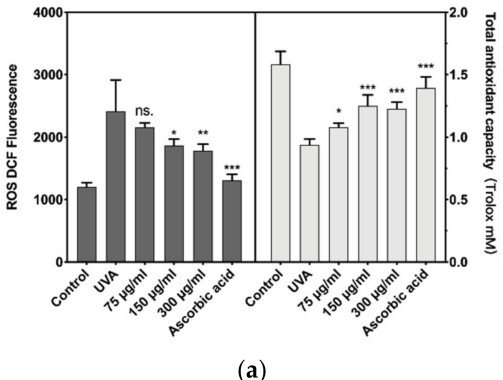

(**a**)

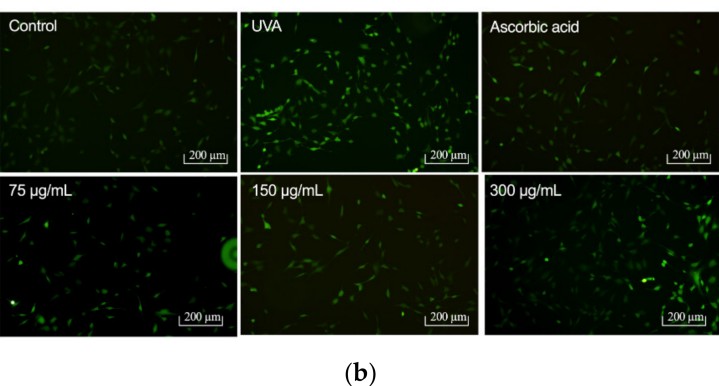

(**b**)

**Figure 3.** Intracellular reactive oxygen species levels and antioxidant capacity. (**a**) The fluorescence intensity of intracellular ROS (Left) and the total antioxidant capacity of the cell (Right), (**b**) the fluorescence photo of intracellular ROS. Difference analysis between sample group and UVA model group: ns.: $p > 0.05$, *: $p < 0.05$, **: $p < 0.01$, ***: $p < 0.001$.

Antioxidant enzymes are vital enzymes for the cellular elimination of reactive oxygen species and the prevention of oxidative stress damage. The antioxidant capacity of cells is directly related to the magnitude of the enzymatic activity of antioxidant enzymes. As shown in Figure 4a–c, the antioxidant enzyme activity and mRNA relative expression of cells decreased after UVA induction. Fortunately, 150 and 300 µg/mL of BAL1 significantly increased the antioxidant enzyme activity and mRNA relative expression level of CCC-ESF-1, especially for catalase (CAT), and the effect was equivalent to that of ascorbic acid. However, the data for 150 and 300 µg/mL BAL1 did not show differences. However, the data for 150 and 300 µg/mL BAL1 did not show differences.

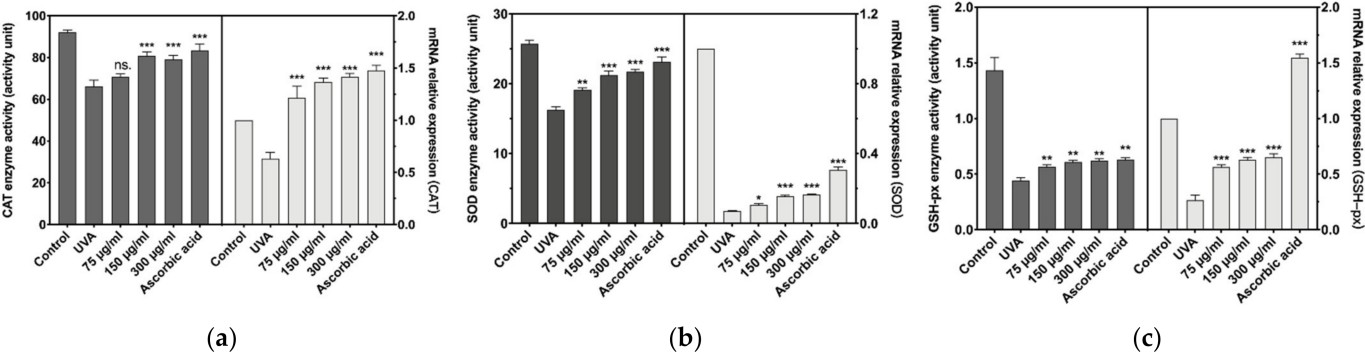

(a)　　　　　　　　　　　　　　　(b)　　　　　　　　　　　　　　　(c)

**Figure 4.** Changes of major intracellular antioxidant enzymes. (**a**) The enzyme activity and transcription level of catalase (CAT). (**b**) The enzyme activity and transcription level of Superoxide dismutase (SOD). (**c**) The enzyme activity and transcription level of glutathione peroxidase (GSH-px). Difference analysis between sample group and UVA model group: ns.: $p > 0.05$, *: $p < 0.05$, **: $p < 0.01$, ***: $p < 0.001$.

### 3.5. Regulation of Nrf2/Keap1 Signaling Pathway by BAL1

Nrf2 binds to the negative regulator Keap1, which induces Nrf2 ubiquitination and proteasomal degradation, blocking the allosteric transfer of Nrf2 to the nucleus. As shown in Figure 5a–c, in the experiments where CCC-ESF-1 was stimulated by UVA, a significant gap was found between cytoplasmic and nuclear Nrf2 content, and the protein content of Keap1 was elevated, indicating that UVA stimulation promoted the binding of Keap1 to Nrf2 and blocked the nuclear ectopic of Nrf2, thus reducing the transcription level of antioxidant enzyme genes downstream of Nrf2, which was consistent with the results in Figure 5d–f. Fortunately, BAL1 has a highly significant effect on Nrf2 signaling ($p < 0.001$), driving nuclear ectopic Nrf2 mainly by blocking the binding of Nrf2 to Keap1. The nuclear displacement ratio (N/C) increased from 0.365 to 1.849, approaching the blank cell state group. Consequently, Heme Oxygenase 1 (HO1), NAD(P)H Quinone Dehydrogenase 1 (NQO1), and Glutamate-Cysteine Ligase Catalytic Subunit (GCLC) were extremely significantly increased in mRNA expression and protein content ($p < 0.001$). These results suggest that BAL1 achieves its antioxidant stress effect mainly by regulating the transduction level of Nrf2.

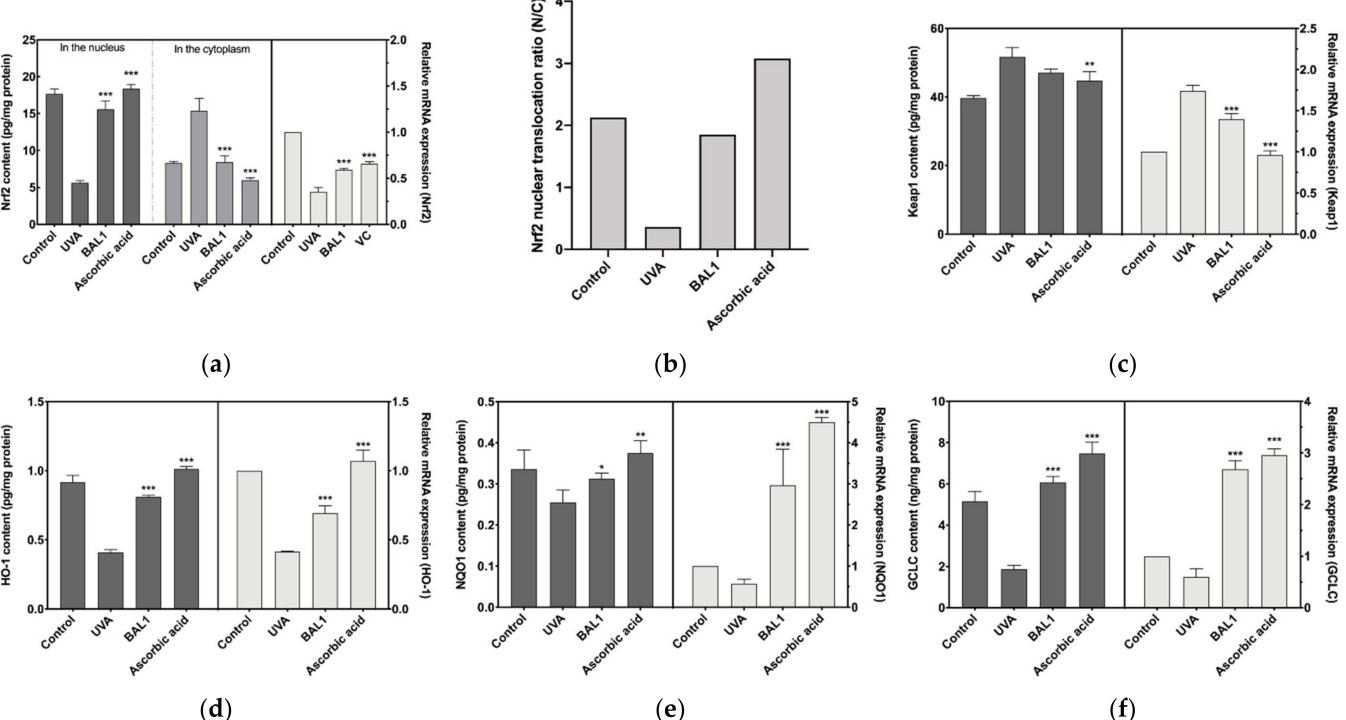

(a)　　　　(b)　　　　(c)

(d)　　　　(e)　　　　(f)

**Figure 5.** Effects of BAL1 on the conduction level of Nrf2/Keap1 signaling pathway. (**a**) Nuclear translocation of Nrf2 during signal transduction and its mRNA expression level. (**b**) The ratio of nuclear displacement of Nrf2. (**c**) The protein content of Keap1 and its mRNA expression level. (**d**) The protein content and mRNA expression level of the downstream regulatory gene HO-1. (**e**) The protein content and mRNA expression level of the downstream regulatory gene NQO1. (**f**) The protein content and mRNA expression level of the downstream regulatory gene GCLC. Difference analysis between sample group and UVA model group: *: $p < 0.05$, **: $p < 0.01$, ***: $p < 0.001$.

### 3.6. Regulation of COL-I Metabolism by BAL1

The mRNA expression level and protein content of MMP-1 were significantly elevated after UVA stimulation of CCC-ESF-1, and the expression level of Smad7 was also considerably elevated. This shows that the elevated level of MMP1 promoted the catabolism of COL-I, while the negative regulation of TGF-β by Samd7 decreased the mRNA expression of COL-I. As shown in Figure 6, BAL1 promotes the repair of cells damaged by UVA irradiation mainly by increasing the mRNA expression of TGF-β and Smad2/3, and reducing the mRNA expression of Smad7. Moreover, to a certain extent, it also reduces the content of MMP-1 to avoid the degradation of COL-I.

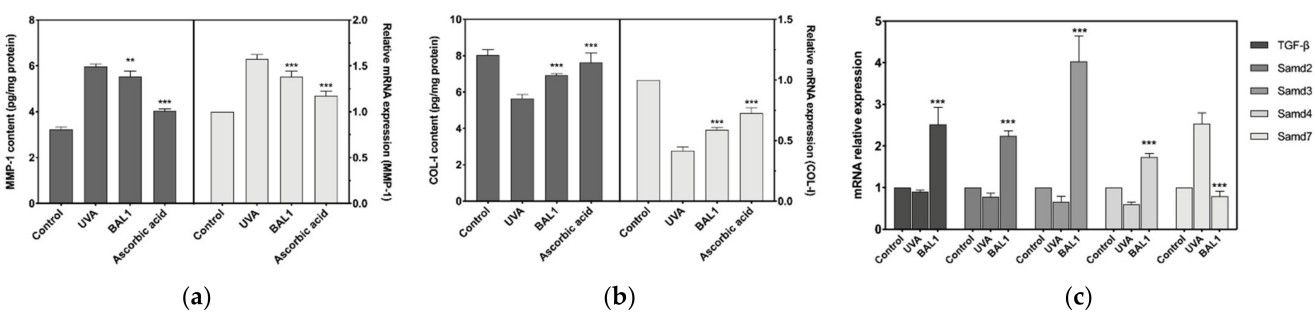

(a)　　　　(b)　　　　(c)

**Figure 6.** Effects of BAL1 on the breakdown and transcription of type I collagen. (**a**) The protein content of MMP1 and its transcription level. (**b**) The protein content of COL-I and its transcription level. (**c**) The protein content and transcription level of TGF-β and its mediating protein Smads. Difference analysis between sample group and UVA model group: **: $p < 0.01$, ***: $p < 0.001$.

## 4. Discussion

Skin photoaging and photooxidative stress refer to the degeneration or damage of skin tissue structure and function caused by long-term overexposure to ultraviolet rays. Long-term exposure to UVA (320–420 nm) can cause damage to the dermal cells of the skin and degradation of the extracellular matrix, resulting in skin folds and relaxation [18]. UVA is the main external factor causing oxidative stress damage to the skin. Increased levels of reactive oxygen species (ROS) in response to UVA stimulation lead to lipid peroxidation, damage to organelles and DNA, and ultimately cell aging or death [19,20]. The transcription factor NF-E2-related factor 2 (Nrf2) translocates from the cytoplasm to the nucleus by dissociating from Kelch-like ECH-associated protein 1 (Keap1) and binds to the ARE to activate downstream antioxidant enzyme gene expression [21,22]. Therefore, Nrf2 is a master regulator of antioxidant defense and a proposed "master regulator" of the aging process [23]. The extracellular matrix (ECM) is a three-dimensional network of collagen and elastin fibers surrounded by a matrix such as hyaluronic acid (HA) to maintain skin plumpness and elasticity. In contrast, matrix metalloproteinases (MMPs) represent a family of extracellular zinc-dependent enzymes whose primary function is to degrade the ECM. An imbalance between them leads to loss of skin structure, resulting in crepey, dry and inelastic aging skin [24]. Exposure to UVA radiation induces matrix metalloproteinase activity to accelerate the degradation of extracellular matrix collagen and elastin fibers [25]. Therefore, the inhibition of MMPs effectively protects the dermal matrix structure, avoids tissue damage, and delays wrinkle formation. In addition, exposure to UV radiation also blocks transforming growth factor-β (TGF-β)/Smad signaling. It reduces type I procollagen synthesis [26], a major pathway for repairing UVA damage.

Previous studies have found that cell-free filtrates, extracellular polysaccharides and extracellular polypeptides of *B. amyloliquefaciens* have strong antioxidant activity [11,14,27,28]. However, no study has been conducted to prove whether its lysate is functional. Our preliminary experiments found that BAL1 obtained by ultrasonic cell crusher contains many small molecular proteins and polypeptides predicted to have antioxidant properties. Through in vitro antioxidant experiments, it was found that BAL1 has strong antioxidant capacity, which is shown in DPPH·, HO·, and ABTS·+ scavenging ability. Moreover, BAL1 has very low or no toxicity to CCC-ESF-1 up to 600 µg/mL and exhibits exceptionally significant protection against cell death due to UVA damage only above 150 µg/mL. To maintain physiological redox homeostasis, endogenous antioxidant enzymes (CAT), SOD, and GSH-px scavenge excess reactive oxygen species from the cells [29]. As shown in Figure 3, 150 µg/mL of BAL1 significantly reduced the intracellular ROS levels, as reflected by ROS's significantly weaker green fluorescence than the UVA model group.

Moreover, the total antioxidant capacity measured by CCC-ESF-1 cell lysate was significantly increased compared with the UVA model group. Its effect was close to that of ascorbic acid, which was attributed to the increased intracellular antioxidant enzyme activity and mRNA expression level of BAL1. The Nrf2 signaling pathway regulated the increased antioxidant enzyme activity and mRNA expression levels, and the binding of Keap1 to Nrf2 could reduce the transduction of the Nrf2 signaling pathway [30]. As shown in Figure 5, the experimental results confirmed that BAL1 promoted the nuclear displacement of intracellular Nrf2. The content and expression level of intracellular Keap1 were significantly reduced compared with the UVA model group. The nuclear displacement ratio (N/C) of Nrf2 increased from 0.365 to 1.849, thus activating the expression of antioxidant enzyme target genes of Nrf2. As we measured, the mRNA expression levels of SOD, CAT, GSH-px, HO-1, NQO1, and CGLC were significantly up-regulated, further reducing the excess intracellular ROS. Further investigation revealed that BAL1 could effectively reduce MMP1 activity and mRNA expression levels, avoiding the degradation of COL-I due to excessive MMP1. Besides, Smads are regulators of TGF-β signaling, in which Smad2/3 is phosphorylated by the TGF-β receptor, which then polymerizes with Smad4 and continues signaling, but Smad7 acts as an inhibitor that blocks the transmission of smad2/3 [31,32]. The regulation of BAL1 in TGF-β/Smad signaling is extremely significant. As shown in

Figure 6, BAL1 promoted the mRNA expression of TGF-β, smad2, smad3, and smad4, and the increase of COL-I content and the up-regulation of mRNA expression level verified that the TGF-β/Smad pathway was promoted.

In summary, the anti-photoaging effect of BAL1 may be closely related to biochemical indicators, Nrf2/Keap1 signaling pathway, and TGF-β/Smad signaling pathway in photoaging skin tissues. We can roughly speculate the potential mechanism of anti-photoaging of BAL1 based on the results obtained from the experiments. As shown in Figure 7, prolonged UVA irradiation leads to a large production of ROS, which then leads to oxidative stress and elevated levels of matrix metalloproteinase enzyme activity and mRNA expression in fibroblasts, followed by the disruption of Nrf2/Keap1 and TGF-β/Smad signaling pathways, impaired cellular function, and degradation of the extracellular matrix. In contrast, BAL1 could promote the nuclear displacement of Nrf2 to activate the transcription of antioxidant enzyme genes to a certain extent, thus reducing the cellular damage caused by oxidative stress due to UVA. It also upregulated the mRNA expression of the transducer and target gene type I pre-collagen in the TGF-β/Samd pathway, which slowed down the degradation of COL-I by MMP-1 to some extent. This suggests that BAL1 can effectively reduce UV light damage to human skin in daily life, thus supporting the significance of its application.

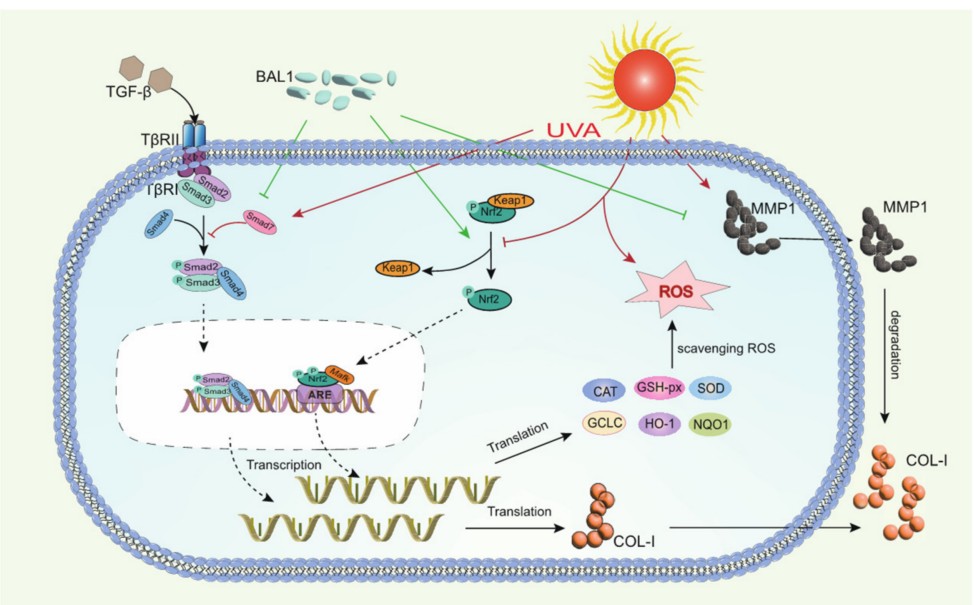

**Figure 7.** Mechanistic diagram of the regulation of NRF2/Keap1 and TGF-β/Smad by BAL1 in human skin fibroblasts.

## 5. Conclusions

In conclusion, our study shows that the lysate of Bacillus amyloliquefaciens BA-1 strain (BAL1) has antioxidant activity and anti-aging effects. Furthermore, BAL1 can enhance the oxidative stress response and enhance the defense against photoaging in skin fibroblasts. This effect may be achieved by regulating Nrf2/Keap1, TGF-β/Samd, and related genes. This study supports the potential application of BAL1 as an antioxidant and anti-aging postbiotic component. In the research, development, and application of epibiotics, BAL1 has shown good efficacy in anti-oxidation and anti-extracellular matrix loss, which fully indicates that the application of BAL1 is a promising method to resist UV damage.

**Supplementary Materials:** The following supporting information can be downloaded at: https://www.mdpi.com/article/10.3390/app12189151/s1, Table S1: Primer Sequence; Table S2: Composition of proteins and peptides in BAL1.

**Author Contributions:** Writing—original draft preparation, Y.Z. and M.L.; visualization and software, Y.Z., J.Z. and Y.J.; writing—review and editing, M.L. and C.W.; methodology and resources, D.W. and D.Z.; funding acquisition, M.L. and C.W. All authors have read and agreed to the published version of the manuscript.

**Funding:** This research was funded by the Research Foundation for Youth Scholars of Beijing Technology and Business University (Grant No. QNJJ2020-04, Funder: Meng Li).

**Institutional Review Board Statement:** Not applicable.

**Informed Consent Statement:** Not applicable.

**Data Availability Statement:** Not applicable.

**Acknowledgments:** Thanks to Meng Li and Changtao Wang for their funding and guidance in the experiment, and also to Chunyu Chen, Hao Fu and Yuzhi Zhang for their helpful inspiration in the experiment and writing.

**Conflicts of Interest:** The authors declare no conflict of interest.

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
