# Peer review of "Bacillus amyloliquefaciens Lysate Ameliorates Photoaging of Human Skin Fibroblasts through NRF2/KEAP1 and TGF-β/SMAD Signaling Pathways"

_applsci, doi:10.3390/app12189151_

Round 1

Reviewer 1 Report

In the article titled ‘Bacillus amyloliquefaciens lysate ameliorates photoaging of human skin fibroblasts through NRF2/KEAP1 and TGF-β/SMAD signaling pathways, the authors used 16 J/cm2 15 UVA stimulation of human embryonic 16 fibroblasts (CCC-ESF-1) to establish a UVA photodamage model to investigate the 17 anti-oxidative and anti-photoaging efficacy of BAL1 and its mechanism of action. This work is really very interesting. However, I have concerns regarding improving the manuscript which I think should be addressed.

1. This manuscript is well written but there are some typos and grammatical errors throughout the whole manuscript which should be revised and checked carefully. The abstract should accurately convey the main point.

2. Introduction part of the paper is weak. The current mechanism of B. amyloliquefacien is not up to date which should be updated till date.

3. Should add some recent related work in the Introduction and discussion sections.

4. The authors should add a subsection ‘summary of the statistical analysis’ in the method sections.

5. The authors are encouraged to include the workflow diagram.

6. The Figures should be placed in where their corresponding contents are presented.

7. The implication of this work and what is the benefits are not defined properly.

8. The authors are encouraged to mention the limitation and future direction of this work.

Author Response

Detailed response to reviewer 1’ comments

August 7, 2022

Journal: Applied Sciences

Manuscript Number: applsci-1880152

Title: “Bacillus amyloliquefaciens lysate ameliorates photoaging of human skin fibroblasts through NRF2/KEAP1 and TGF-β/SMAD signaling pathways”

Authors: Yongtao Zhang, Jingsha Zhao, YanBing Jiang, Dongdong Wang, Dan Zhao, Changtao Wang, Meng Li.

Dear Editor and reviewer,

Thank you very much for giving us this opportunity to revise our manuscript. Your comments and suggestions on our manuscript have been encouraging and helpful. The following pages provide a detailed point-by-point response to your comments. Note that the reviewer’ comments are presented in Italics, and our responses are in Roman and blue font. In addition, we addressed all these major points and other issues carefully and revised the manuscript accordingly, and we highlight the revised parts in the article in yellow. Please let me know if you have any further questions.

In response to Reviewer 1's suggestion, we have made the following modifications and responded:

  1. This manuscript is well written but there are some typos and grammatical errors throughout the whole manuscript which should be revised and checked carefully. The abstract should accurately convey the main point.

Reply:Thanks to the reviewer's suggestion, we have carefully reviewed and revised the manuscript for typos and grammatical errors during the revision process, and have also made reasonable corrections regarding the unclear main ideas expressed in the abstract.

  1. Introduction part of the paper is weak. The current mechanism of B. amyloliquefacien is not up to date which should be updated till date.

Reply:Thanks to the reviewer's question. Since the research on B. amyloliquefaciens mainly focuss on agriculture and the sideline industryThere are few related studies, so it is not introduced to the latest research in the first draft, which we have revised and added in the introduction section.

  1. Should add some recent related work in the Introduction and discussion sections.

Reply:Thanks to the reviewer's suggestion, as with the previous question, we have revised and added to the introduction and discussion sections.

  1. The authors should add a subsection ‘summary of the statistical analysis’ in the method sections.
    Reply:Thanks to the reviewer's suggestion, we have made appropriate modifications in the summary "2.10 Statistical analysis" in the manuscript.
  2. The authors are encouraged to include the workflow diagram.

Reply: Thanks for the reviewer's suggestion. I have inserted figures in the discussion section of the article to make it easier to understand the study.

  1. The Figures should be placed in where their corresponding contents are presented.

Reply: Thanks to the reviewer's suggestion, we have placed each image in the manuscript in the corresponding summary section.

  1. “The implication of this work and what is the benefits are not defined properly.” and “The authors are encouraged to mention the limitation and future direction of this work”

Reply:Thanks to the reviewer's suggestion, the definition, direction and significance of the research work in the manuscript were not well presented in the introduction, discussion and conclusion sections, so we have made reasonable additions in the abstract, introduction, discussion and conclusion sections.

Sincerely,

Meng Li

Beijing Key Lab of Plant Resource Research and Development, Beijing Technology and Business University, Fucheng Road, Beijing 100048, China

Tel.: +86-13426015179

Reviewer 2 Report

Comment and suggestion to authors:

Manuscript ID: applsci-1880152

Type: Article

Titled:  Bacillus amyloliquefaciens lysate ameliorates photoaging of human skin fibroblasts through NRF2/KEAP1 and TGF-β/SMAD signaling pathways

1) For the antioxidant ability of BAL1, why did the authors studied only 1 assay using DPPH? Because there are many antioxidant assays to investigate the different antioxidant mechanisms, and many studied employed DPPH with the other assays to determine the antioxidant mechanisms. This information should be clearly provided in the main manuscript as well.

2) The greater number of another related published works should be added to discuss with the results from the section “3.1 LC-MS/MS analysis”, “3.2 Antioxidant capacity of BAL1”, “3.4 The effect of BAL1 on cellular antioxidant capacity”, “3.5 Regulation of Nrf2/Keap1 signaling pathway by BAL1” and “3.6 Regulation of COL-I metabolism by BAL1

3) As the authors wrote that “In this study, we used 16 J/cm 2 UVA stimulation of human embryonic fibroblasts (CCC-ESF-1) to establish a UVA photodamage model to investigate the anti-oxidative and anti-photoaging efficacy of BAL1 and its mechanism of action”. The additional figure to show the graphic of anti-oxidative and anti-photoaging mechanisms of BAL1 should be prepared and added to the main manuscript to give an overview of its anti-oxidative and anti-photoaging mechanisms for the readers.

4) It would be better to add 1-2 sentences of future perspective to the section “Conclusions” for the better understanding of the readers.

5)There are some spelling mistakes and grammatical error found in this manuscript.

Author Response

Detailed response to reviewer 2’ comments

August 7, 2022

Journal: Applied Sciences

Manuscript Number: applsci-1880152

Title: “Bacillus amyloliquefaciens lysate ameliorates photoaging of human skin fibroblasts through NRF2/KEAP1 and TGF-β/SMAD signaling pathways”

Authors: Yongtao Zhang, Jingsha Zhao, YanBing Jiang, Dongdong Wang, Dan Zhao, Changtao Wang, Meng Li.

Dear Editor and reviewer,

Thank you very much for giving us this opportunity to revise our manuscript. Your comments and suggestions on our manuscript have been encouraging and helpful. The following pages provide a detailed point-by-point response to your comments. Note that the reviewer’ comments are presented in Italics, and our responses are in Roman and blue font. In addition, we addressed all these major points and other issues carefully and revised the manuscript accordingly, and We highlight the revised parts in the article in yellow. Please let me know if you have any further questions.

In response to Reviewer 2's suggestion, we have made the following modifications and responded:

  1. For the antioxidant ability of BAL1, why did the authors studied only 1 assay using DPPH? Because there are many antioxidant assays to investigate the different antioxidant mechanisms, and many studied employed DPPH with the other assays to determine the antioxidant mechanisms. This information should be clearly provided in the main manuscript as well.

Reply:Thanks to the reviewer's suggestion, we added the Fenton assay to detect the scavenging ability of BAL1 for hydroxyl radicals and also the ABTS assay to detect the scavenging ability of BAL1 for ABTS·+ to increase the exploration of the scavenging ability of BAL1 for different types of free radicals.

  1. The greater number of another related published works should be added to discuss with the results from the section “3.1 LC-MS/MS analysis”, “3.2 Antioxidant capacity of BAL1”, “3.4 The effect of BAL1 on cellular antioxidant capacity”, “3.5 Regulation of Nrf2/Keap1 signaling pathway by BAL1” and “3.6 Regulation of COL-I metabolism by BAL1”.

Reply:Thanks to the reviewer's suggestion, we cite the relevant studies in "3.1 LC-MS/MS analysis", and we present and discuss the other parts in the Discussion section.

  1. As the authors wrote that “In this study, we used 16 J/cm 2 UVA stimulation of human embryonic fibroblasts (CCC-ESF-1) to establish a UVA photodamage model to investigate the anti-oxidative and anti-photoaging efficacy of BAL1 and its mechanism of action”. The additional figure to show the graphic of anti-oxidative and anti-photoaging mechanisms of BAL1 should be prepared and added to the main manuscript to give an overview of its anti-oxidative and anti-photoaging mechanisms for the readers.

Reply:Thanks to the reviewer's suggestion, We have inserted a ‘Graphic abstract’ in the discussion section of the manuscript to summarize better and understand the research content of this paper.

  1. It would be better to add 1-2 sentences of future perspective to the section “Conclusions” for the better understanding of the readers.
    Reply:Thanks to the reviewer's suggestion, the future perspective of the research work in the manuscript was not well presented in the discussion and conclusion sections, so we have made reasonable additions in the discussion and conclusion sections.
  2. There are some spelling mistakes and grammatical error found in this manuscript.

Reply: Thanks to the reviewer's suggestion, we have carefully reviewed and revised the manuscript for typos and grammatical errors during the revision process.

Sincerely,

Meng Li

Beijing Key Lab of Plant Resource Research and Development, Beijing Technology and Business University, Fucheng Road, Beijing 100048, China

Tel.: +86-13426015179

Round 2

Reviewer 1 Report

No comments